# Impacts of COVID-19 Pandemic on Sleep Quality Evaluated by Wrist Actigraphy: A Systematic Review

**DOI:** 10.3390/jcm12031182

**Published:** 2023-02-02

**Authors:** Luiz Felipe Ferreira-Souza, Marize Julianelli-Peçanha, Ana Carolina Coelho-Oliveira, Christianne Martins Corrêa da Silva Bahia, Laisa Liane Paineiras-Domingos, Aline Reis-Silva, Márcia Cristina Moura-Fernandes, Luiza Carla Trindade-Gusmão, Redha Taiar, Danubia da Cunha Sá-Caputo, Amandine Rapin, Mario Bernardo-Filho

**Affiliations:** 1Laboratório de Vibrações Mecânicas e Práticas Integrativas—LAVIMPI, Departamento de Biofísica e Biometria, Instituto de Biologia Roberto Alcântara Gomes and Policlínica Piquet Carneiro, Universidade do Estado do Rio de Janeiro, Rio de Janeiro 20950-003, RJ, Brazil; 2Programa de Pós-Graduação em Saúde, Medicina Laboratorial e Tecnologia Forense, Universidade do Estado do Rio de Janeiro, Rio de Janeiro 20950-003, RJ, Brazil; 3Coordenação Médica do Hospital Estadual da Mulher Heloneida Studart, São João de Meriti 25565-171, RJ, Brazil; 4Programa de Pós-Graduação em Fisiopatologia Clínica e Experimental, Faculdade de Ciências Médicas, Universidade do Estado do Rio de Janeiro, Rio de Janeiro 20550-900, RJ, Brazil; 5Serviço de Neurologia, Setor de Distúrbios do Sono, Hospital Universitário Pedro Ernesto, Universidade do Estado do Rio de Janeiro, Rio de Janeiro 20550-031, RJ, Brazil; 6Programa de Pós-Graduação em Ciências Médicas, Faculdade de Ciências Médicas, Universidade do Estado do Rio de Janeiro, Rio de Janeiro 20551-030, RJ, Brazil; 7Departamento de Fisioterapia, Instituto de Ciências da Saúde, Universidade Federal da Bahia, Salvador 40110-902, BA, Brazil; 8MATériaux et Ingénierie Mécanique (MATIM), Université de Reims Champagne-Ardenne, 51100 Reims, France; 9Faculté de Médecine, Université de Reims Champagne Ardennes, UR 3797 VieFra, 51097 Reims, France

**Keywords:** sleep, social isolation, actigraphy, COVID-19 pandemic, circadian rhythm, sleep schedules

## Abstract

COVID-19 has probably contributed as a risk factor for sleep disturbance. Actigraphy has been used to evaluate sleep complaints in self-isolated populations and frontline doctors during the COVID-19 pandemic. This systematic review aims to summarize the impact of the COVID-19 pandemic on sleep through wrist actigraphy, estimating sleep latency, total sleep time, awakening-after-sleep onset, and sleep efficiency. Searches were conducted of observational studies on the PubMed, Embase, Scopus, Web of Science, and PEDro databases from 1 December 2019 to 31 December 2022. Ninety articles were found, and given the eligibility criteria, fifteen were selected. Six studies were classified by the National Health and Medical Research Council as evidence level IV, two studies as level III-3, and seven studies as level III-2. According to the ACROBAT-NRSI instrument, three studies were classified as having a “serious” risk of bias, two as having “critical” risk, four as having “moderate” risk, and six as having “low” risk. In the selected publications, various populations were evaluated via actigraphy during the COVID-19 pandemic, with reports of “poor” sleep quality. Actigraphy may be a relevant tool to assess individual day–night rhythms and provide recommendations under enduring pandemic conditions. Moreover, as actigraphy presents objective data for sleep evaluations, it is suggested that this method be used in similar pandemics and that actigraphy be included as part of the sleep hygiene strategy.

## 1. Introduction

Coronavirus disease (COVID-19) causes social and psychological consequences that, directly and indirectly, might impact mental health during the pandemic and in the future [1]. As a respiratory disease, COVID-19 has affected the services that care for the needs of people with mental health problems around the world [2,3]. These mental health problems can affect regular lifestyles related to the health behaviors and daily activities of individuals of all ages, including restrictions on their free-living physical activities. In addition, increased sedentary time, sleep fragmentation, and consequent effects on the quality of life have been verified [4], together with the important rise in psychological distress and signs of mental illness in the general population [5]. Mak et al. [6] observed a similar negative impact on the mental health of those who face sudden events that culminate in post-traumatic stress disorder symptomatology. Such traumatic events, including those caused by the COVID-19 outbreak, can produce psychological distress and anxiety symptoms that negatively affect sleep quality [7].

Sleep is a biological process that is essential for life and optimal health. Normal healthy sleep is characterized by sufficient duration, good quality, appropriate timing and regularity, and the absence of sleep disturbances and disorders [8]. Sleep plays a fundamental role in the regulation of emotions, since an alteration in its pattern may have direct consequences on the emotional functioning of an individual the day after such an alteration occurs [9]. These changes may diminish the amount of time spent sleeping and staying in bed and affect the positive associations for home, relaxation time, leisure activities, and sleep, resulting in higher levels of stress [10]. 

Sleep disorders usually manifest in one of three ways: failure to obtain the necessary amount or quality of sleep (sleep deprivation), inability to maintain sleep continuity (interrupted sleep, also called sleep fragmentation, difficulty in maintaining sleep, and middle insomnia), and events that occur during sleep (sleep apnea, restless legs syndrome, bruxism) [11,12]. Lack of adequate sleep is a major source of many harmful diseases related to the heart, the brain, psychological changes, high blood pressure, diabetes, and/or weight gain. Forty to 50% of the world’s population suffers from poor or inadequate sleep [13]. Studies on social isolation and its effects on both psychological well-being and sleep quality have pointed out the occurrence of several problems, including decreased exposure to sunlight, dietary changes, room temperature changes due to confinement, reduced social interaction, overwork under stressful conditions, and living with constant uncertainty and insecurity regarding health status [10,14].

Despite public awareness, anxiety levels affect sleep quality during epidemics, including periods of home quarantine, lockdown, self-isolation, and/or social distancing; specific occupations may be particularly affected [15]. Such increases in anxiety levels occurred during the COVID-19 pandemic. Moreover, it was observed during the COVID-19 pandemic that a broad spectrum of clinical manifestations led to psychological distress due to social isolation, financial risk, and unemployment [16,17]. Among the types of psychological suffering, pandemic-related sleep disorders have received much attention from public and mental health professionals worldwide [18,19]. There is evidence that before the COVID-19 pandemic, health care personnel did not feel that good sleep quality had important effects on health [20], although epidemiological studies indicated a strong association between the circadian rhythms and human health [21]. Circadian rhythms, which take place in 24 h daily cycles, can be affected not only by a person’s internal clock, but also by external factors, such as social rhythms, daily schedules, and daylight exposure, which can lead to delay in the biological clock, as sunlight has been considered an important zeitgeber [22]. Throughout the COVID-19 pandemic, analyses of cloud-based activity records have shown relevant sleep duration and bedtime changes [23]. In studies developed during the pandemic, the sleep patterns of participants were assessed by subjective measures, and the Pittsburgh sleep quality index (PSQI) was frequently used, providing evidence of sleep disturbances [24,25,26]. Although the PSQI questionnaires provide clinically relevant indicators of sleep, previous research has shown divergent results between self-reported and actigraphy-assessed sleep [27], and future research should seek to validate existing results using more objective techniques. 

Remote patient monitoring (RPM) technologies are essential for researchers and clinicians. Wearable actigraphy monitors allow for a more objective, but simultaneously ecological, assessment of sleep [28]. The use of simple, non-invasive activity monitors (actigraphy) can provide important clinical information and activity levels during the post-acute rehabilitation management of patients with sleep disorders [29]. Subjective sleep reports are usually confounded and inaccurate. In addition, patients with insomnia tend to overestimate or underestimate sleep length, compared with actigraphy-defined sleep duration [30]. Although polysomnography (PSG) is considered the gold standard method of measuring sleep, it is a complementary test that requires an intrusive and expensive assessment of sleep indices. Wrist actigraphy emerged as an alternative tool in implementing a method of effortless application for several consecutive days [31]. In comparison to PSG, it is a low-cost, non-intrusive instrument that can be utilized to estimate sleep quality and quantity. It has up to 93% accuracy in healthy adults in determining total sleep time and sleep efficiency [32]. 

Actigraphy may be the most efficient measurement tool for assessing individual daytime and nighttime rhythms and providing recommendations during long-lasting pandemic conditions by estimating sleep latency, total sleep time, awakening-after-sleep onset, and sleep efficiency [33]. It has also been claimed that actigraphy measures can be used to estimate sleep amounts and sleep continuity in patients with sleep disorders [34]. Connected-health technologies permit the monitoring of combined findings and facilitate remote clinical trials, leading to a reduced potential for spreading COVID-19 [35]. Quantitative measures via PSG or actigraphy provide useful information that is complementary to self-reported sleep data [36,37]. Studies using surveys and sleep-related applications on mobile devices have suggested that the pandemic has contributed to increases in sleep disruption or the onset of new sleep disturbances and concluded that individuals had delayed sleep-wake schedules, increases in total sleep time, decreases in sleep efficiency, and less sleep quality [38,39,40]. 

An understanding of the psychological and physiological effects of the COVID-19 restrictions on sleep time perception may aid in the development of evidence-driven strategies, now and in the future for emerging public health matters [30]. Bin Heyat et al. [41] emphasized the importance of a theranostics approach, adding different herbal medicines to other procedures to improve the quality of human life. Future research should explore ways to improve restorative sleep during pandemics using more objective techniques, such as actigraphy [28], as there is a research gap in this area. Furthermore, the research question considered in this article is whether sleep quality could be affected during the COVID-19 pandemic. In light of the disturbances in sleep during outbreaks of infectious disease, the current systematic review aims to summarize the effects on sleep quality during the period of the COVID-19 pandemic, through data acquired by wrist actigraphy.

## 2. Materials and Methods

### 2.1. Research Question

The purpose of the current systematic review was to answer the following question: Can sleep quality be affected during the COVID-19 pandemic? The PECOS components (P = problem, E = exposition, C = comparison, O = outcomes, S = study design) were the major considerations in addressing that research question [42]. In this article, P = the sleep quality during the COVID-19 pandemic; E = individuals in specific conditions during COVID-19 pandemic; C = not applicable; O = negative impacts on sleep quality by actigraphy; S = observational studies.

### 2.2. Search Method

The search was performed using the following electronic online databanks: PubMed, Embase, Web of Science, Scopus, and the Physiotherapy Evidence Database (PEDro), It was conducted from 1 December 2019 to 31 December 2022. The keywords “COVID-19* and sleep and social isolation and circadian rhythm and actigraphy” and “Clinical trial or Randomized controlled trial (RCT)” were used to find publications. In situations in which it was impossible to conduct an RCT (for operational, ethical, or financial reasons), observational studies were presented as an alternative.

### 2.3. Registration

Research at the International Prospective Registry of Systematic Reviews (PROSPERO) [43] was conducted before developing the current systematic review to exclude the existence of systematic reviews that were carried out for the same purpose. As no similar study was found, the protocol of the current systematic review was registered (23 January 2022) with PROSPERO (www.crd.york.ac.UK/Prospero/, 23 January 2022) under CRD 42022306220. The search and analysis were carried out following Preferred Reporting Items for Systematic Reviews and Meta-Analysis (PRISMA) guidelines. A detailed protocol can be accessed online at PROSPERO.

### 2.4. Operational Definition

Actigraphy is a simple method of detecting sleep and circadian rhythm disorders. The data of objective parameters for evaluating sleep disorders can be obtained with an electronic actigraph worn on the wrist, ankle, or waist. An example is shown in Figure 1. Detailed tracking of data acquired by the actigraph is critical, as it contributes to the prediction, prevention, and personalized evidence-based targeted analysis of individual patient profiles, including light exposure, duration, intensity of activity–rest cycles, and movement using an accelerometer [44]. A wrist actigraph is a non-invasive monitor of human rest-activity cycles [45,46] and produces estimates that are highly correlated with polysomnography [47]. In general, actigraphy is accomplished via sleep logs and is utilized for measuring total sleep time (TST), sleep latency (SL), sleep efficiency (SE, corresponding to TST and time in bed (TIB)), and wake-after-sleep onset (WASO). The accuracy and reliability of modern actigraphs have increased due to the improvement of lithium batteries, piezoelectric motion sensors, and extended memory storage capabilities [48]. Applications used as middleware perform the interface between the actigraph and the computer to record, analyze, and score physical activity and sleep/wake characteristics, facilitating the estimation and even the prevention of sleep-related diseases [49].

### 2.5. Study Selection and Data Extraction

Independently of the year of publication, all the publications found on the searched five databases were exported to a file, and two authors (L.F.F-S. and M.B-F.) manually removed the duplicates. After that, three steps were considered in the current review. Records were identified in the databases (identification), and two reviewers (M.J-P. and M.C.M-F.) independently assessed the titles and abstracts and excluded irrelevant publications based on the eligibility criteria (screening). Full texts that were selected were analyzed according to the eligibility criteria, and all relevant publications were considered in this current systematic review (included). One publication that was identified from its website was included in the current systematic review. If the two reviewers did not agree, the matter was solved by a third reviewer (ACCO). Gray literature was not considered in the current systematic review.

The researchers (M.J-P. and M.C.M-F.) carried out the data extraction for author and year, demographics (country, age, gender), population (sample size), specific conditions, and wrist actigraphy.

### 2.6. Eligibility Criteria

The publications included in the current systematic review assessed the effects of the COVID-19 pandemic on the sleep quality of individuals exposed, or not exposed, to quarantine, via actigraphy; were fully published in English; and constituted a cross-sectional design, a cohort study, or a control case.

Duplicate publications were excluded, as well as letters, comments, conference abstracts, book chapters, books, incomplete publications, systematic reviews, narrative reviews, or meta-analyses. In addition, publications that did not address sleep quality or actigraphy, or that did not specifically describe findings related to COVID-19, were rejected. Publications that associated other diseases and comorbidities with sleep quality were also excluded.

### 2.7. Level of Evidence

The level of evidence (LE) of the selected publications followed standards of the National Health and Medical Research Council (NHMRC) 2003–2009 [50], and the hierarchy of evidence was used to classify the included studies in the current systematic review. Six levels were applied: (a) LE I—systematic review of level II studies; (b) LE II—RCT; (c) LE III-1—pseudorandomized controlled trial; (d) LE III-2—comparative study with concurrent controls (non-RCT, case-control study, cohort study, interrupted time series without a parallel control group); (e) LE III-3—comparative study without concurrent controls (historical control, interrupted time series without a parallel control group, two or more single-arm studies); (f) LE IV—case series with either post-test or pre-test/post-test outcomes.

### 2.8. Risk of Bias

The risk of bias in the included publications was assessed with the “A Cochrane Risk of Bias Assessment Tool for Non-randomized Studies” instrument (ACROBAT-NRSI) [51], which compares the health effects of interventions. ACROBAT-NRSI includes seven domains at intervention, pre-intervention and post-intervention, and post-intervention. Each item is classified as a moderate, low, serious, or critical risk of bias. Moreover, when there is no information, that fact is noted. Using checklists, two independent authors (L.F.-S. and A.C.C-O.) assessed the risk of bias. Concordance between the reviewers was estimated using k-statistics [52] data to review the full text and assess the relativity and risk of bias. In case of discordance, the obtained data was assessed by a third reviewer (L.L.P-D.), who made the final decision. The k rate of concordance between the reviewers’ findings was k = 0.84.

### 2.9. Effect Measures

Synthesis Without Meta-Analysis (SWiM) [53] was conducted because the outcome measures were too diverse to yield different study designs, with various populations, various sleep problems, and individuals in specific conditions that were not due to the COVID-19 pandemic. However, the current study summarized the findings in a systematic review.

### 2.10. Bibliographic Research

Figure 2 shows a PRISMA flowchart [54] with the steps of the bibliographic research used in this systematic review, identifying the number of the selected publications and the full search process. Eighty-nine articles were found in the databases, fifty-seven of which were not considered because they were reviews (systematic reviews or metanalysis or narrative reviews), duplicates, and/or studies that did not correlate with the subject. Of the remaining thirty-two articles, eighteen were not considered because they were not about actigraphy, sleep quality, and/or COVID-19. In addition, conference abstracts, incomplete articles, publications with other comorbidities, and publications in a language other than English were excluded. One article was selected from the Google search engine and included as eligible at the end of the flowchart. Therefore, based on the eligibility criteria, fifteen studies met all the requirements of this study.

## 3. Results

Considering the LE (NHMRC), six studies were classified as Level IV [55,56,57,58,59,60], two studies were classified as Level III-3 [38,61] and seven studies were classified as Level III-2 [30,36,62,63,64,65,66]. The evaluated studies did not present interventions in the investigations. Regarding the risk of bias, according to the ACROBAT-NRSI instrument, three publications were classified as having a “serious” risk of bias [55,56,59], two publications were classified as having a “critical” risk [38,61], four publications were classified as having a “moderate” risk [57,58,60,66], and six publications were classified as having a “low” risk [30,36,62,63,64,65]. The multiple comparisons of the stratified analyses interfered with the power characteristics of the studies, due to social isolation. In Figure 3, it is possible to see the graphic visualization of the risk of bias in the evaluated studies (robvis) [67].

Table 1 shows the characteristics of the populations that were subject to the selected articles. The populations were from China [30,36], the United States [38,58,61,63], Australia [65], Spain [60,62], Canada [64], India [56], Brazil [57,59], and Italy [55,66]. A total of 714 individuals took part in the studies; 326 were females and 318 were males. Two studies did not define gender [59,61]. Although the study by Giulia and Sreedharan [56] presented only one female evaluated by the actigraph, it was included in the study because it was a longitudinal pilot study focused on a specific postmenopausal condition. The ages of the individuals ranged from 56.4 ± 10.8 months of age to 67 years. Of the fifteen studies that were selected, eleven publications were about comorbidities during the COVID-19 pandemic associated with individuals in various specific conditions (studying online at home, multiple sclerosis, post-menopausal condition, and type 1 narcolepsy) [30,36,38,55,56,57,58,62,63,64,65]. Two publications were about patients treated in hospital [60,66]. Only one study addressed medical performance during the pandemic [61], and another study was about pilots flying humanitarian missions during COVID-19 [59].

Table 2 shows the use of the actigraph according to the population involved and presents the sleep-disturbance results. Wrist actigraphy was used in all publications to assess sleep quality. The specific models included the following: Actiwatch (1 and 2), wGT3X-BT, ActTrust, GENEActiv, Zulu watch, Somnowatch plus, and Micro Motiongger Watch. The time of use of the devices in the selected studies ranged from five days to more than six months. The analysis of the recorded data was defined according to the following categories: (i) sleep duration, defined as the total time between the onset of sleep (bedtime) until the start of the day’s activity (wake time) on overall-average monitoring nights; (ii) sleep efficiency (%),calculated as the total interval between the time spent sleeping and the time spent in bed (min), showing actual sleep time (min); (iii) estimated sleep continuity (and possible sleep disturbances). The actual waking time (min), the number of awakenings, and the average time of each awakening (min) were evaluated. The results of the selected studies, their aims, and their conclusions are reported in Table 2.

## 4. Discussion

Studies showed that survivors of Severe Acute Respiratory Syndrome (SARS) [68], Middle East Respiratory Syndrome (MERS) [69], and influenza A (H1N1) [70] had neuropsychiatric effects with adverse impacts that persisted for a long time, even after the epidemic ended, showing symptoms of anxiety, depression, stress, and post-traumatic stress disorder. The current systematic review comes at a critical time when the world still faces the challenges of the COVID-19 pandemic, which, in the face of established social limits, led to robust changes in the lifestyles of the individuals that contributed to massive sleep deficits [71]. Naturally, the findings might aid in future social disturbances that may be due to many factors. Through the restrictions imposed in many countries due to COVID-19, physical activities decreased significantly, and this could contribute to a worsening in sleep quality [72]. Moreover, Sonza et al. [73] reported that the COVID-19 pandemic changed the physical exercise practice and habits and psychological well-being of individuals in several ways. In assessing sleep quality during the COVID-19 pandemic lockdown, Pinto et al. [74] pointed to sleep difficulties reported by the subjects and not objectively measured by actigraphy or polysomnography, given the imposed social distancing. The evidence indicates that sleep disorders are underestimated and underdiagnosed in primary care [75] and that actigraphy is a sampling technique that can observe these disorders. When comparing the current systematic review using wrist actigraphy in several specific conditions and the results of self-assessment questionnaires [24,25,26], wrist actigraphy obtains more objective data on sleep quality in the context of the COVID-19 pandemic, as self-assessment can allow divergences in the results [75] due to the behaviors, thoughts, and feelings at moments of difficulty and the form used at a distance to collect results via digital media [4,15,76].

Actigraphy is as an alternative tool for the evaluation of individuals with insomnia. It may be performed outside the laboratory environment for several consecutive nights, unlike polysomnography (the gold standard), which is inappropriate in situations of social isolation [77,78] as it requires the presence of the patient in the laboratory; polysomnography is also a high-cost procedure [79]. Furthermore, other psychological factors, such as economic dissatisfaction and the quality of social relationships [80], were observed to affect the self-assessment of sleep time, as insufficient sleep time is associated with higher levels of social support deficiency and depressive symptoms [81]. He et al. [30] used self-reported sleep questionnaires and actigraphy devices to obtain subjective and objective sleep time parameters during COVID-19. It was shown that people who self-isolated at home reported significantly earlier sleep onset and earlier wake times than those whose sleep was measured by actigraphy, which suggested that participants tended to overestimate their sleep schedules. Therefore, actigraphy is capable of defining sleep patterns that are characteristic of sleep apnea and periodic leg movements. It has been used extensively in intervention studies tracking patients’ sleep progress over time [82].

Considering the physical and functional parameters of sleep, Andreu-Caravaca et al. [62] found that actigraphy-measured sleep quality diminished with large and moderate impacts on sleep efficiency and sleep time, respectively, after home confinement in people with multiple sclerosis. In addition, Wang et al. [36] showed that levels of physical activity (average activity (count) 1679.66 ± 546.84) and ambient light exposure (average ambient light-lux—37.79 ± 47.24) were low in individuals who stayed home during the pandemic, which may have affected their mental and physical health. A study by Peterson et al. [38] also found evidence of increased sleep disruption after lockdown; the findings were consistent with previous studies based on self-reported data, which observed delayed wake times and decreased sleep quality after confinement. Lokhandwala et al. [63] reported that students in remote learning environments during the pandemic slept longer, compared to their sleep patterns before the pandemic, and emphasized that actigraphy was permitted to measure children’s sleep objectively rather than via parental reporting measures. Corroborating these findings, Stone et al. [65] reported on remote learning in the face of lockdown, and found that adolescents slept more, slept longer, and slept later, suggesting that interventions in the scheduling of school activities to increase sleep duration—probably in the face of greater psychological suffering related to reduced social relationships—decreased exposure to daylight in the morning and the use of electronic equipment without due moderation [57].

Filardi et al. [55] observed that children and adolescents with type 1 narcolepsy also delayed their sleep phase and slept more during the day during the lockdown. Eldringhoff et al. [61] provided a preliminary examination of the collected data, broadly suggesting that medical personnel who present with symptoms of infection may have difficulty in obtaining adequate sleep, a fact confirmed by Ferini-Strambi et al. [20] who previously indicated that poor sleep quality among medical teams is prevalent and increased during the COVID-19 pandemic. Reasons for this include long working hours, risk of infection, shortages of protective equipment, loneliness, physical fatigue, and separation from families [83]. In evaluating pilots during humanitarian missions by actigraphy, Devine et al. [59] demonstrated how important it is to obtain adequate rest when a risk of fatigue is posed, not only during periods of active duty but also during downtime, supporting the care we must have for certain professional classes with respect to sleep quality during pandemics. Conte et al. [66] showed that the detrimental impacts on sleep verified by the initial pandemic outbreak have not decreased throughout the subsequent waves of contagion, as indicated by the continuing decrease in sleep quality.

Benítez et al. [60] highlighted the importance of considering sleep and circadian health in patients surviving COVID-19 after hospital discharge from an intensive care unit (ITU), when they had a greater fragmentation of the rest–activity rhythm. According to the conclusions of the selected publications, and based on the various findings via actigraphy, poor sleep quality was observed in individuals during the COVID-19 pandemic. There is evidence of increased sleep disturbance [38]; daytime activity levels and light exposure were negatively affected [36]; perceptions of sleep onset and wake times were affected [30]; and people slept more during the daytime and napped more frequently [55,64]. These findings might be due to isolation, which resulted in inconsistent sleep parameters, irregular circadian rhythm, and decreased exposure to daylight in the morning [57]. After analyzing the included studies and considering their limitations, the results suggest that actigraphy may be a better objective sleep measurement tool than subjective perceptions or parental reports, providing essential information for the medical care of patients with sleep disorders [84].

Actigraphy has become an essential tool in sleep medicine and is increasingly used in the clinical care of patients with sleep and circadian rhythm abnormalities [33]. However, Danzig et al. [85] reported that there are substantial limitations in the estimation of sleep time and sleep efficiency in individuals with sleep disorders when applied in clinical practice, due to substantial diagnostic errors. This demonstrates that these devices are not very accurate in outperforming existing diagnostic tools. Several factors were responsible for clinically significant sleep disturbances during the pandemic, in association with comorbidities common to COVID-19, such as diabetes, cardiovascular diseases, asthma, obesity, hypertension, and chronic obstructive pulmonary disease, which can worsen patients’ clinical conditions [86]. There is an urgent need for collaboration between sleep specialists, engineers, and device manufacturers to yield improved sleep-tracking devices that are accurate enough to apply to individual patient decision-making across various sleep and medical disorders [85].

Given the Danzig report, if pandemic-induced home confinement is necessary, physical home-training programs would need to be implemented to mitigate the problems associated with sedentary living, as well as to achieve benefits from the physiological and psychological effects of exercise. Medical support during major disasters should be strengthened and potentially delivered through telemedicine, as this comprehensive approach could reduce psychological distress and improve sleep quality [75]. Some emergency measures can improve the sleep quality of frontline health workers during the pandemic: implementing exercise programs, making psychological counseling available, using treatment strategies with drug intervention, if necessary [87], providing individual interventions for mental health services [88], establishing a shift system to allow rotation of rest, and using a hotline for telephone guidance [89]. Clear communication, limiting shifts, providing rest areas, broadening access to detailed rules on the use and management of protective equipment, and specialized treatment training for patients with COVID-19 are also part of these measures [90]. For confined individuals, it is suggested that public awareness be raised through information measures, that psychosocial intervention be carried out to improve social isolation [91], that emergency psychological intervention be available, that many harmful problems be avoided, and that regular exercise be maintained [15,92]; for patients undergoing treatment in isolation, progressive muscle relaxation is suggested as an auxiliary method [93].

Sleep disturbances can run a chronic course and there is a need to educate people about sleep disturbances during the pandemic. Sleep disorders can be better evaluated using technology to obtain more accurate results and a better understanding of the effects of social isolation in populations that are subject to a pandemic. These results can be useful at the individual and government policy levels in generating proper pandemic response strategies.

### 4.1. Future Directions

The COVID-19 pandemic altered the mental health of the population during periods of social isolation, and sleep disorders were in evidence. Moreover, a society’s commitment to mental health might include consideration of sleep disturbance during the post-COVID period [60]. In this context, the use of actigraphy as an ideal method for estimating sleep efficiency was highlighted, as it appears to provide for the collection of data that can be used remotely and for a prolonged period without causing discomfort to patients. As sleep loss could contribute to various physiological impairments in human beings and impact the immune system, actigraphy is proposed as part of the sleep hygiene strategy [94]. Furthermore, a bibliometric analysis, i.e., a rigorous method for assessing robust volumes of scientific and popular information about actigraphy, will be performed.

#### 4.1.1. Strength of the Study

The main strength of this systematic review was to describe the relevance of actigraphy focused on the assessment of sleep problems in periods of social isolation, rather than using self-assessment questionnaires and polysomnography. In future studies, it is expected that actigraphy will be added as an adjunct to other forms of assessment that already exist, to allow for more objective results in the assessment of sleep problems and to obtain more reliable results in similar situations of social isolation. Thus, it is of extreme importance to implement public health policies that include assistance protocols concerning individual and collective mental health in conjunction with pandemic response strategies during and after the event.

#### 4.1.2. Limitations of the Study

The current systematic review has some limitations and the results of the current systematic review must be evaluated with caution. While five databases were utilized, the inclusion of additional sources of data could improve the number of publications that were included in this review. Moreover, the following considerations should be noted: different models of actigraphs were used; self-selection bias was possible in light of the imposed social distance, which could have inhibited the results of the study; the exclusion of non-English language studies may have led to the omission of relevant information; more representative research for other affected countries is needed; the cross-sectional design of this study did not identify a causal relationship among the variables and this must be better explored in randomized controlled trials; and the sample could be expanded to include greater diversity in terms of gender, various types of situations, and different evaluation times.

## 5. Conclusions

This study aimed to summarize the effects on sleep quality of individuals during the COVID-19 pandemic, using data obtained by wrist actigraphy. Considering the overall accuracy in measuring, wrist actigraphy may be the best instrument for obtaining data about the patient and providing suggestions for improving behavioral patterns. Given the impact of sleep-related lifestyle changes during the pandemic, future studies with objective and standardized distance-assessment methods may be used to obtain more effective data.

## Figures and Tables

**Figure 1 jcm-12-01182-f001:**
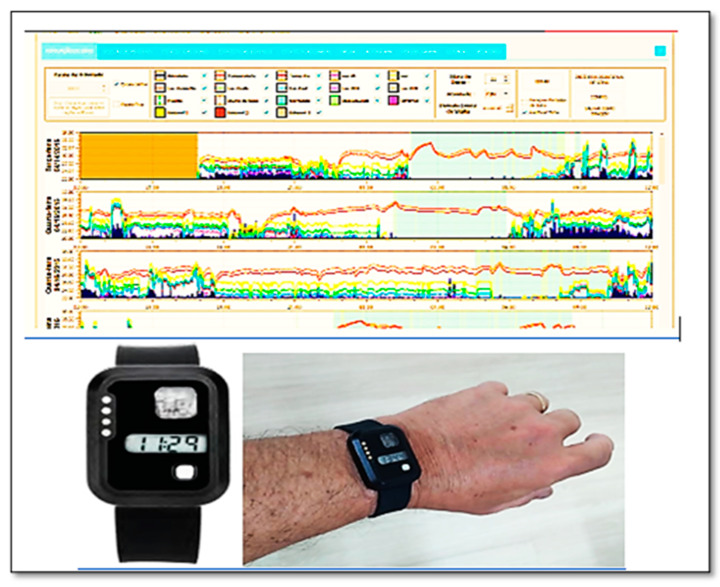
Wrist actigraph (ActTrust-2) on the left hand and sleep score graph display showing captured objective parameter data.

**Figure 2 jcm-12-01182-f002:**
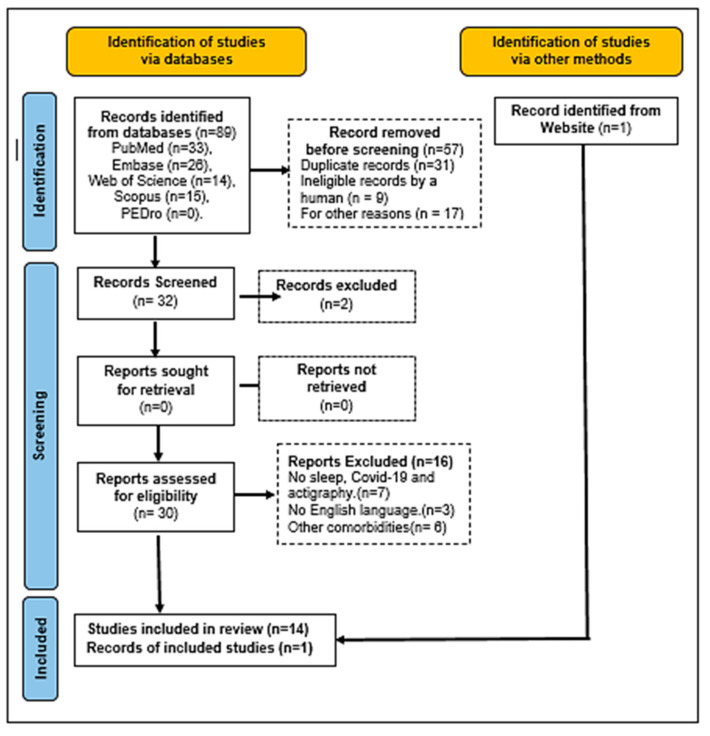
PRISMA 2020 flowchart of bibliographic research and the various steps in the selection of the articles.

**Figure 3 jcm-12-01182-f003:**
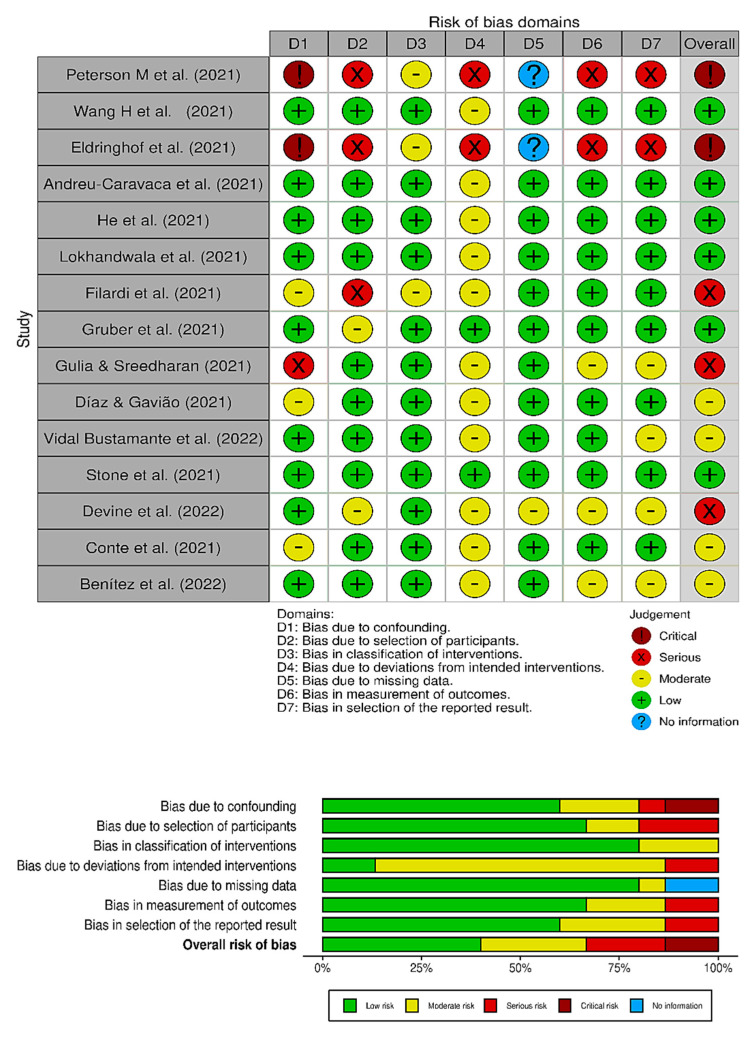
ACROBAT-NRSI risk of bias of the selected publications [30,36,38,55,56,57,58,59,60,61,62,63,64,65,66].

**Table 1 jcm-12-01182-t001:** Individual characteristics inf the selected studies regarding the author, year, country, population size, gender, age, and specific condition during COVID-19.

Author (Year)	Country	Population Size	Gender (Female/Male)	Age (Years or M_age_ ± SD) * or Median (25–75) **	Specific Condition
Peterson et al. (2021) [38]	USA	9 individuals in stay-at-home order	5/4	From 22 to 48	in-home confinement
Wang et al. (2021^)^ [36]	China	70 individuals in stay-at-home-order	46/24	From 31 to 60	in-home confinement
Eldringhof et al. (2021) [61]	USA	50 medical personnel working through the pandemic	Not reported	35.15 ± 9.97	working in the pandemic.
Andreu-Caravaca et al. (2021) [62]	Spain	17 individuals with multiple sclerosis	10/7	43.50 ± 11.23	in-home confinement
He et al. (2021) [30]	China	70 participants into home stayers and area-restricted workers	46/24	From 31 to 65	isolation at home or working during COVID-19
Lokhandwala et al. (2021) [63]	USA	16 preschool-aged children	3/13	56.4 ± 10.8 months	in online learning remote programs or not
Filardi et al. (2021) [55]	Italy	18 type 1 narcolepsy children and adolescents	7/11	14.44 ± 2.01	in-home confinement
Gruber et al. (2021) [64]	Canada	62 adolescents	44/18	From 12 to 16	in a domestic environment during COVID-19
Gulia & Sreedharan (2021) [56]	India	1 postmenopausal woman	1/0	56	during COVID lockdown period
Dias & Gavião, (2022) [57]	Brazil	19 students	10/9	From 19 to 33	during social isolation in COVID-19 pandemic
Vidal Bustamante et al. (2022) [58]	USA	43 university students	25/24	From 18 to 19	studying during the COVID-19 pandemic
Stone et al. (2021) [65]	Australia	59 students	33/26	12.8 ± 0.4	during COVID-19 pandemic
Devine et al. (2022) [59]	Brazil	20 pilots flying across five humanitarian missions	Not reported	Over 18	in humanitarian missions during the COVID-19 pandemic in ULR
Conte et al. (2021) [66]	Italy	82 patients	40/42	32.5 ± 11.5	during distinct phases of the COVID pandemic emergency
Benítez et al. (2022) [60]	Spain	172 patients	56/116	61 (52.8–67.0)	three months after hospital discharge during COVID-19

* Mage: average age; SD: standard deviation. ** Median (25th–75th percentile) USA—United States of America. ULR: ultra-long-range operations.

**Table 2 jcm-12-01182-t002:** Description of the selected studies and the sleep evaluation by actigraph.

Author	Wrist Actigraph	Sleep Monitoring Time	Aim	Result	Conclusion
Peterson et al. [38]	Actiwatch-2 (Philips Respironics)	Two weeks	Measure the sleep disturbances during stay-at-home orders	Delayed sleep onset by 53.4 ± 15.1 min (F[1,101] = 12.46, *p* < 0.001), delayed final awakening by 104.3 ± 19.6 min (F[1,101] = 28.43, *p* < 0.001), longer sleep duration (F[1,101] = 6.06, *p* = 0.016), increased number of awakenings (F[1,101] = 13.00, *p* < 0.001), trend for increased intermittent wakefulness (F[1,101] = 3.88, *p* = 0.052) post lockdown.	Evidence of increased sleep disruption. Observed later wake times and decreases in sleep quality.
Wang et al. [36]	wGT3X-BT	Five consecutive days and nights	assess effects of night-time sleep and daytime activity on well-being	Sleep onset time M (23:50) SD (1:25) range (22:29–5:21); wake-up time M (8:02) SD (1:26) range (4:52–12:39); sleep midpoint M (3:56) SD (1:19) range (1:11–9:00); TST M (8:13) SD (1:9) range (5:46–13:6).	Sleep, daytime activity levels, and light exposure were negatively affected by the COVID-19 pandemic.
Eldringhof et al. [61]	Actiwatch Philips.	Over six months	Understand the effects of sleep on the severity of COVID-19 symptoms in medical personnel	Poorer sleep quality t(255.59) = 5.78, *p* =< 0.001,poorer mood upon waking t(258.03) = 6.53, *p* =< 0.001, feeling less alert upon waking t(255.61) = 4.56, *p* =< 0.001, and spending more time awake at night t(266.98) = −7.29, *p* =< 0.001.	Medical personnel with symptoms of infection may have trouble sleeping properly.
Andreu-Caravaca et al. [62]	wGT3X-BT	Measured in two moments (pre- and post-home confinement of 5 days each)	Analyze the effect of home confinement on sleep quality in people with MS	Sleep quality decreased with significant and moderate effects on sleep efficiency (ES = 1.27, *p* = 0.01) and sleep time (ES = 0.81, *p* = 0.01), CI 95%.	Worsening of sleep quality variables was seen in people with MS.
He et al. [30]	wGT3X-BT	Five consecutive days, one participant for three days	Assess the impact of COVID-19 lockdown on sleep time perception	A significant difference in wake-up times [t(68) = 2.576, *p* = 0.012, d = 0.625] and sleep onset times [t(68) = 2.513, *p* = 0.014, d = 0.609], however not in TST.	Home isolation affected participants’ perception of sleep onset and wake-up time, but not their perceived total sleep time.
Lokhandwala et al. [63]	Actiwatch Spectrum Plus (Philips Respironics)	Sixteen consecutive days and nights	Investigate children’s previous sleep patterns with coping during COVID-19	Longer sleep duration in remote students who slept 44 min more during the night [t_(14)_ = 2.92, *p* = 0.01, d = 1.46].	Remote students slept significantly more through the night than students not involved in home learning.
Filardi et al. [55]	Micro Motionlogger Watch, Ambulatory Monitoring	Fourteen consecutive days	Evaluate the effects of lockdown COVID-19 on nighttime sleep and daytime naps of NT1 children and adolescents	Lower levels of DMA (t(17) = 5.75, *p*< 0.0001), higher and DTST (t(17) = −2.59, *p* < 0.05) and increased frequency of naps (t(17) = −2.89, *p* = 0.01), while the mean nap duration was not changed (t(17) = −1.46, *p* = ns).	NT1 children and adolescents slept more during the daytime and napped more frequently during the lockdown.
Gruber et al. [64]	Actiwatch (AW-64 series; Mini-Mitter)	Seven consecutive nights	Compare the sleep patterns of adolescents and behaviors immediately before and during COVID-19 pandemic	Average bedtime was 1:28 h later, average wake-up time was 2:13 h later than their pre-pandemic schedules (F(1.61) = 68.55, *p* < 0.000, η_p_^2^ = 0.53 and F(1.61) = 94.33, *p* < 0.000, η_p_^2^ = 0.61, respectively. Daytime sleepiness was lower during the pandemic compared to pre-pandemic (F(1.61) = 13.17, *p* < 0.01, η_p_^2^ = 0.18. Sleep quality did not change between the two time points. Sleep duration was 1:01 h longer during the pandemic compared to the pre-pandemic (F(1.47) = 14.68, *p* < 0.000, η_p_^2^ = 0.19).	Sleep schedule was delayed and sleep duration was longer in the period of the COVID-19 pandemic compared to pre-pandemic sleep.
Gulia & Sreedharan. [56]	Somnowatch plus (Somnomedics)	Four weeks without intervention; twenty-four weeks with interventions (yoga-hydra and walking exercise)	Investigate yoga-hydra and walking intervention as a strategy to induce improved sleep and well-being in the elderly population during COVID lockdown	Pre-intervention values of 4 weeks sleep latency (score), as M ± SD (2.07 ± 0.55) and post-intervention of 24 weeks (1.24 ± 0.58), *p* < 0.001. TST pre-intervention ((h:min),as M ± SD (6.09 ± 1.38) and post-intervention of 24 weeks (6.53 ± 1.34), *p* < 0.05.	Regular practice of yoga-hydra with walking exercise improved the feeling of satisfying sleep and the sleep latency.
Dias & Gavião. [57]	ActTrust^®^ (model AT0503 Condor Instruments)	24 h period for seven consecutive days	Investigate nocturnal sleep parameters, estimate the activity-rest pattern, and determine the exposure to light with actigraphy.	Two clusters were formed: normal sleepers (n = 13) and short sleepers (n = 6). Circadian function index. Normal sleepers M ± SD (0.44 ± 0.08), short sleepers M ± SD (0.43 ± 0.08) *p* < 0.05. Sleep parameters M ± SD. Normal sleepers-bedtime (h:min)—(01:33 ± 10.560), Get up time (h:min)—(08:59 ± 1.88), TIB (h:min)—(8.19 ± 1.11), sleep period (h:min)—(7.18 ± 1.10), sleep efficiency (%)—(93.26 ± 4.80); short sleepers—bedtime (h:min)—(03:04 ± 7.03), get-up time (h:min)—(09:09 ± 2.09), TIB (h:min)—(6.46 ± 1.32), sleep period (h:min)—(5.40 ± 1.41), sleep efficiency (%)—(89.43 ± 7.22) *p* < 0.05. Normal sleepers had significant higher exposure to daylight (U = 37.00; *p* = 0.015) than short sleepers.	During social isolation presented inconsistent sleep parameters, irregular circadian rhythm and decreased exposure to daylight during the morning.
Vidal Bustamante et al. [58]	GENEActiv Original, Activinsights Ltd.	Three-month remote monitoring during the third year of college during the COVID-19 pandemic	Assess affective and behavioral experiences associated with mental health and sleep outcomes in students during the COVID-19 pandemic	Daily actigraphy per subject: range (132–249), M (220), med (227), SD (24.44); sleep duration (r): range (0.28–16.05), M (7.28), med (7.26), SD (0.54); sleep timing regularity index: range (0–1), M (0.75), Med (0.76), SD (0.05).	For most students, academic stress was common. Important psychological distress was emphasized for stressors involving social relationships.
Stone et al. [65]	GENEActiv (original, Activinsights)	For one to two weeks during in-person learning and during remote learning	Examine whether COVID-19 induced change in a school mode (in-person versus remote learning) was associated with changes in sleep, circadian timing, and mood in early adolescents.	During remote learning on average went to sleep 26 min later and woke 49 min later, compared to in-person school days. Sleep duration was longer in remote learning. In-person learning—sleep onset time (h:min), M ± SD (22:16 0:40), Wake time (h:min), M ± SD (7:12 0:34), sleep duration (h:min), M ± SD (8:55 0:35); remote learning—sleep onset time (h:min), M ± SD (22:44 1:03), wake time (h:min), M ± SD (7:55 0:41), sleep duration (h:min), M ± SD (9:10 0:45).	During remote learning in the face of lockdown, adolescents slept more, and less self-reported anxiety, slept longer/later, and were more in line with their circadian rhythms.
Devine et al. [59]	Zulu watch, (Institutes for Behavior Resources)	Five round-trip flights from humanitarian missions (Brazil/China/Brazil), with a duration of 30 flight hours each	Evaluate the ability of the SAFTE-FAST 4.0 AutoSleep function to predict pilot sleep duration throughout the missions compared to subjective (sleep diary) and objective (Zulu watch) measures of sleep during the airline’s COVID-19 humanitarian missions	Comparison of average in-flight sleep duration by flight segment: AutoSleep predicted 235 ± 20 min, compared to the 325 ± 128 min reported by the sleep diary, or the 246 ± 132 min recorded by Zulu watches. Paired samples *t*-tests showed that diary reports were higher than AutoSleep predictions (t = 6.05, df = 151, *p* ≤ 0.001) or Zulu watch sleep duration (t = 3.73, df = 150, *p* ≤ 0.001). AutoSleep predictions of sleep duration were not significantly different from Zulu watch sleep duration during FDPs (t = 0.69, df = 151, *p* = 0.48).	It is important to get adequate rest not only during periods of active duty but also during downtime. Giving up sleep during layovers may pose a risk of fatigue during ULR. however, the time zone difference must be considered.
Conte et al. [66]	Motionlogger^®^ Microwatches	Two nights of recording during weekdays	Compare findings in the third wave with data collected during previous pandemic waves	No significant differences were found in any actigraphy sleep parameter between the 2 nights of recording bedtime (h:min), M ± SD (00:33 ± 1), wake time (h:min), M ± SD 08:33 ± 1:22), sleep midpoint (h:min), M ± SD (04:36 ± 1:21), frequency of awakenings ≥ 1 min/TST(h), M ±SD (1.78 ± 0.94). TIB and WASO (%) are M ± SD (8.09 ± 1.10 h) and (6.71% ± 5.82%), respectively.	Unfavorable effects on sleep verified by the initial pandemic outbreak have not decreased throughout the subsequent waves of contagion.
Benítez et al. [60]	Actiwatch 2 (Philips Respironics)	Seven days	To assess the sleep and circadian rest-activity pattern of critical survivors of COVID-19 three months after discharge from the hospital	Objective evaluation of sleep (actigraphy), TST (h), med (25~75) 6.98 (6.33–7.67), TIB (h), 8.38 (7.73–9.10), sleep efficiency (%), 84.6 (81.0–88.3), latency (min), 10.0 (5.00–18.0), WASO (min), 51.0 (39.0–66.0).	Critical survivors of COVID-19 may present bad sleep quality and modifications in the circadian rest–activity pattern three months after discharge from the hospital.

DMA—mean activity counts during daytime; DTST—estimated diurnal total sleep time; NT1—type 1 narcolepsy; MS—multiple sclerosis; CI: confidence interval; AutoSleep: sleep-prediction algorithm; URL: ultra-long-range operations; FDPs: flight duty periods; DST—daylight saving time; M ± SD—mean and standard deviation; WASO: wake-after-sleep onset; TST: total sleep time; TIB: time in bed; %: percent; h: hour; min: minute; *p*: p value; Med: median.

## Data Availability

The findings of this study are relevant to the evaluation of the quality of sleep in pandemic situations. Moreover, information about a simple and safe device, the actigraph, is presented.

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
