# Peer review of "Impacts of COVID-19 Pandemic on Sleep Quality Evaluated by Wrist Actigraphy: A Systematic Review"

_jcm, 2023, doi:10.3390/jcm12031182_

Round 1

Reviewer 1 Report

This is an interesting systematic review considering people's sleep and Covid-19. The subject is of interest to clinicians and those involved in public health. It restricts itself to sleep quality as assessed with actigraphy, which is reasonable.

However, the paper could be improved somewhat by having consistency in the wording throughout the manuscript. The title "The effects of COVID-19 on sleep quality ..." suggests that the study will be looking at the illness itself and how it affects sleep quality. But in the Introduction it is stated that the paper is about the effects of the pandemic on sleep quality, which is much broader suggesting that not only the illness itself, but other effects of lock downs, stress etc would be considered. In the Materials and Methods section it specifically states that the "general population" will be the focus, and that there will be a comparison between people in quarantine and those not subjected to quarantine.

In the Results section I was thus looking for the comparisons between those in quarantine and those not in quarantine in the studies that they reviewed. I could not find this comparison. It seems from reading the submitted manuscript that the studies reviewed were simply describing sleep quality in people in a variety of situations related to the Covid-19 pandemic from home confinement, during lock downs to pilots on flights. This is all interesting, but such descriptions do not address the authors' stated aims.

The details of the actigraphy parameters that were reported in the studies need to be made much clearer.

The Discussion would be easier if it was broken into paragraphs, perhaps with sub-headings, rather than being one long paragraph.

Author Response

Reviewer 1

This is an interesting systematic review considering people's sleep and Covid-19. The subject is of interest to clinicians and those involved in public health. It restricts itself to sleep quality as assessed with actigraphy, which is reasonable.

Authors: We thank you, and we agree about the relevance of the subject.

However, the paper could be improved somewhat by having consistency in the wording throughout the manuscript. The title "The effects of COVID-19 on sleep quality ..." suggests that the study will be looking at the illness itself and how it affects sleep quality.

Authors: We agree and we changed the title to “Impacts of COVID-19 pandemic on sleep quality evaluated by wrist actigraphy: A systematic review.”

But in the Introduction it is stated that the paper is about the effects of the pandemic on sleep quality, which is much broader suggesting that not only the illness itself, but other effects of lock downs, stress etc would be considered.

Authors: We agree and we thank you. As we changed the title, in this revised version, we connected the Introduction with the title of the manuscript. Moreover, we have also followed the suggestion of the reviewer 2, and we modified the paragraphs and some statements presented in the introduction.

In the Materials and Methods section it specifically states that the "general population" will be the focus, and that there will be a comparison between people in quarantine and those not subjected to quarantine.

Authors: We thank you and we agree. Following your comments, we adopted the PECOS to a better comprehension of our manuscript.

The purpose of the current systematic review was to answer the following question: Can sleep quality be affected during the pandemic Covid-19? The PECOS (P=Problem, E=Exposition, C=Comparison, O=Outcomes, S= Study design ) are the major components of the research question [42]. P= the sleep quality during the Covid-19 pandemic; E= Individuals in specific conditions during Covid-19 pandemic; C= Not applicable; O= Negative impacts on sleep quality by actigraphy; S= Observational studies.

In the Results section I was thus looking for the comparisons between those in quarantine and those not in quarantine in the studies that they reviewed. I could not find this comparison. It seems from reading the submitted manuscript that the studies reviewed were simply describing sleep quality in people in a variety of situations related to the Covid-19 pandemic from home confinement, during lock downs to pilots on flights.

Authors: We thank you and we agree. As we modified in the Material and Methods, it was clarified that no comparison was did.

This is all interesting, but such descriptions do not address the authors' stated aims.

The details of the actigraphy parameters that were reported in the studies need to be made much clearer.

Authors: we agree and we thank you. We added a legend at the bottom of the tables with the description as you requested.  

The Discussion would be easier if it was broken into paragraphs, perhaps with sub-headings, rather than being one long paragraph.

Authors: we thank you and we agree. We have divided in several paragraphs to clarify the understanding of the Discussion.

Reviewer 2 Report

Dear Author, 

Author Response

Reviewer 2

  • This article is one type of systematic review, please correct the submission. Because it showed a

research article.

Authors: We asked the academic Editor to correct the type of the submission to “systematic review”

  • In the abstract, no need for a structured abstract. Please remove the headings such as the objective,

methodology, results, and conclusion.

Authors: thank you. We followed you and we changed .

  • You only collect the data between Mar 19th,2022, and Aug 15th, 2022. My suggestion is please collect the data between 1 December 2019 to 31 December 2022. It’s very important to improve

this article.

Authors: We changed and we collected the publications from December 1st,2019 to December 31st, 2022. It was verified no alteration in the number of the selected publications.

  • Whenever you will write the XX et al. so please use citations together. For eg: XX et al. [no]... .

Authors: Thank you. We did this throughout the manuscript.

  • In the introduction, the author should add some interesting work like : # Progress in Detection of

Insomnia Sleep Disorder: A Comprehensive Review. Curr. Drug Targets 2020, 22, 672–684,

doi:10.2174/1389450121666201027125828. # Role of Oxidative Stress and Inflammation in

Insomnia Sleep Disorder and Cardiovascular Diseases: Herbal Antioxidants and

Anti Inflammatory Coupled with Insomnia Detection Using Machine Learning. Curr. Pharm.

Des. 2022, 28, 3618–3636, doi:10.2174/1381612829666221201161636. # Detection, Treatment

Planning, and Genetic Predisposition of Bruxism: A Systematic Mapping Process and Network

Visualization Technique. CNS Neurol. Disord. - Drug Targets 2020, 20, 755–775,

doi:10.2174/1871527319666201110124954.

Authors: We agree, we thank you and we added these publications.

  • The author should add Research Question under the introduction section.

Authors: We agree and we introduce in the Introduction section the sentence: … Furthermore, the research question is to verify if the sleep quality could be affected during the Covid-19 pandemic Covid-19.

  • In the Introduction, the author should add: a summary of the work in the third last paragraph

about the current study, the second last paragraph about the main contributions point-wise, and

the last paragraph about the structure of the paper.

Authors: We agree and we thank you for. Following your suggestion, we did a new structure  of the Introduction section to indicate your suggestions.

  • In Section 2, the authors should divide “Research question, registration and search” into

sub sections. For eg: 2.1 Ethical Registration; 2.2 Search Method.

Authors: We thank you and we agree. We did the alterations following your requests.

  • Please add PRISMA under section 2.

Authors: We thank you, we agree and we moved the PRISMA figure.

  • All figures are blurry, please improve them.

Authors: We thank you, we agree and we improved the quality of the figures.

  • The author should add some interesting figures basically biological.

Authors: We thank you, we agree and we introduced a figure in the Material and Methods section.

  • The author should add Research Gap.

Authors: We thank you, we agree and we introduce in the introduction section the sentence…. as actigraphy [2829], since there is a research gap in this area. Furthermore,…

  • The author should bibliometric analysis if possible.

Authors: We thank you, we agree and we introduce at the end of the Discussion section the sentence…. system, actigraphy is proposed to be included as part of the sleep hygiene strategy [9490]. Furthermore, a bibliometric analysis, that is a popular and rigorous method for assessing robust volumes of scientific information about the actigraphy, will be performed.

  • The authors should compare your review with other review articles related to this work.

Authors: We thank you, we agree and we did in the Discussion section.

Round 2

Reviewer 2 Report

Accepted